# Zero-Shot Video Restoration and Enhancement with Assistance of Video Diffusion Models

## Abstract

Although diffusion-based zero-shot image restoration and enhancement methods have achieved great success, applying them to video restoration or enhancement will lead to severe temporal flickering. In this paper, we propose the first framework that utilizes the rapidly-developed video diffusion model to assist the image-based method in maintaining more temporal consistency for zero-shot video restoration and enhancement. We propose homologous latents fusion, heterogenous latents fusion, and a COT-based fusion ratio strategy to utilize both homologous and heterogenous text-to-video diffusion models to complement the image method. Moreover, we propose temporal-strengthening post-processing to utilize the image-to-video diffusion model to further improve temporal consistency. Our method is training-free and can be applied to any diffusion-based image restoration and enhancement methods. Experimental results demonstrate the superiority of the proposed method.

## 1 Introduction

Recently, Denoising Diffusion Probabilistic Models (DDPMs) Dhariwal & Nichol (2021) have shown advanced generative capabilities on the top of GANs, which inspire people to explore diffusion-based restoration method. Different from using supervised learning and diffusion framework to train model for specific restoration task Saharia et al. (2022), Song & Ermon (2019); Lugmayr et al. (2022); Choi et al. (2021); Kawar et al. (2022); Wang et al. (2022); Chung et al. (2022); Fei et al. (2023); Rout et al. (2024); Chung et al. (2023); He et al. (2023); Rout et al. (2023) utilizes pretrained image diffusion model for universal zero-shot image restoration. Among them, Rout et al. (2024); Chung et al. (2023); He et al. (2023); Rout et al. (2023) works on latent space with text-to-image diffusion model, constrains the content between generated result and degraded images in the reverse diffusion process. PSLD Rout et al. (2024) proposes the first framework to solve zero-shot image restoration with text-to-image latent diffusion model. Chung et al. (2023); He et al. (2023); Rout et al. (2023) proposes different strategies to guide the sampling. But there is no temporal knowledge in pretrained text-to-image diffusion model to process videos. Although these methods has achieved promising results on image restoration, directly applying them to video restoration will result in severe temporal flickering.

Recently, there are some methods focusing on zero-shot video restoration/enhancement by designing training-free temporal modules and insert them into image IR methods. Cao et al. (2025) proposes the short-long-range temporal attention layer, temporal consistency guidance, spatial-temporal noise sharing, and an early stopping sampling strategy for zero-shot video restoration and enhancement. DiffIR2VR Yeh et al. (2024) proposes flow-guided video token merging to improve temporal consistency when applying diffusion-based image restoration models for video restoration. Although these method achieve the improvement of temporal consistency, their temporal modules are also limited by training-free manner. Along with the development of video diffusion model, more and more amazing T2V/I2V model have emerged. These T2V/I2V models have been trained on a large amount of video data to learn powerful temporal priors which can preserve the temporal consistency of results. We propose to utilize the temporal priors in these T2V/I2V models to overcome the limitations of previous training-free temporal modules.

Inspired by FVDM Lu et al. (2024), we propose homologous latents fusion to directly fuse the latents between image restoration/enhancement method and their homologous T2V models. Homologous T2V model should share the same VAE with the image IR method. But the SOTA T2V models with better temporal consistency are all Heterogenous T2V model, their 3D VAE were not used for any image IR method. Their latents can not be directly fused with image IR methods. Therefore, we further propose the heterogenous latents fusion to get rid of the restrictions on T2V models. Through our homologous and heterogenous latents fusion, we can utilize any kind of SOTA T2V model to assist the image restoration/enhancement model in achieving more temporal consistent video restoration/enhancement. But how to set the fusion ratio at different timestep is also a challenge problem. If the fusion ratio is too small, it would not work. Too large fusion ratio will result in strong temporal consistent but blurry results or destroy the structure. Inspired by the development of chain-of-thought (COT) Lightman et al. (2023); Ma et al. (2023); Snell et al. (2024); Guo et al. (2025), we propose the COT-based fusion ratio strategy to solve this problem. We design a process reward model based on video quality as test-time verifiers within CoT reasoning paths. At each timestep, we sample several fusion ratio, choose the best fusion ratio from process reward model. Although fusing the latents between the IR/IE model and the T2V model can lead to better temporal consistency, the severe degree of temporal flickering of the IR model itself will still limit the performance. Therefore, we propose temporal-strengthening post-processing, which utilizes the I2V model to further improve the temporal consistency.

Based on the above observations, we propose a novel framework for zero-shot video restoration and enhancement.

Our contributions are summarized as follows

- First, we propose the first framework for Zero-shot Video Restoration and enhancement with assistance of Video diffusion models (ZVRV).
- Second, we proposed the homologous latents fusion, heterogenous latents fusion and COT-based fusion ratio strategy to utilize both homologous and heterogenous T2V diffusion models to complement the image IR method.
- Extensive experiments demonstrate the effectiveness of our method in achieving temporal-consistent zero-shot video restoration and enhancement.

## 2 RELATED WORK

### 2.1 ZERO-SHOT IR WITH LATENT DIFFUSION MODELS

The success of diffusion generative models enlightened zero-shot image restoration methods, which can be further devided in pixel-space zero-shot IR and latent-space zero-shot IR. Pixel-space zero-shot IR Song & Ermon (2019); Lugmayr et al. (2022); Choi et al. (2021); Kawar et al. (2022); Wang et al. (2022); Chung et al. (2022); Fei et al. (2023) utilizes pretrained unconditional image diffusion models working on pixel-space, like Dhariwal & Nichol (2021). Latent-space zero-shot IR Rout et al. (2024); Chung et al. (2023); He et al. (2023); Rout et al. (2023) utilizes pretrained text-to-image latent diffusion models, like Stable Diffusion Rombach et al. (2022). PSLD Rout et al. (2024) is the first framework to solve zero-shot image restoration with text-to-image latent diffusion model. Chung et al. (2023) proposes a prompt tuning method to jointly optimizes the text embedding in the sampling. He et al. (2023) uses the historical gradient information to guide the sampling. But all these methods are designed for image recovery problems, there exists severe temporal flickering when apply them to degraded videos.

### 2.2 ZERO-SHOT VIDEO EDITING

Along with the development of powerful pre-trained text-to-image diffusion models like Stable Diffusion Rombach et al. (2022), diffusion-based zero-shot video editing Wu et al. (2023b); Zhao et al. (2023); Yang et al. (2023) has gained increasing attention, which utilizes these off-the-shelf text-to-image diffusion model and mainly solve the temporal consistency problem. FateZero Qi et al. (2023) follows Prompt-to-Prompt Hertz et al. (2022) and fuse the attention maps in the DDIM inversion process and generation process to preserve the motion and structure consistency. Text2Video-Zero Khachatryan et al. (2023) proposes cross-frame attention and motion dynamics to

enrich the latent codes for better temporal consistency. FVDM Lu et al. (2024) proposes to fuse the latents between T2I and T2V latents for zero-shot video editing. And FVDM can be regarded as a kind of homologous latents fusion. Different from this work, we transfer the homologous latents fusion for video restoration and enhancement, and further propose the heterogenous latents fusion to get rid of the restrictions on T2V models. Through our homologous and heterogenous latents fusion, we can utilize any kind of SOTA T2V model to assist the image restoration/enhancement model in achieving temporal consistent video restoration/enhancement. In addition, we propose a novel COT-Based Fusion Ratio Strategy to better control the fusion ratio. And we propose temporal-strengthening post-processing to utilize the I2V model to further improve the temporal consistency.

### 2.3 ZERO-SHOT VIDEO RESTORATION AND ENHANCEMENT

ZVRD Cao et al. (2025) proposes the short-long-range temporal attention layer, temporal consistency guidance, spatial-temporal noise sharing, and an early stopping sampling strategy for zero-shot video restoration. DiffIR2VR Yeh et al. (2024) proposes flow-guided video token merging to improve temporal consistency when applying diffusion-based image restoration models for video restoration. Although these methods design training-free temporal modules and insert them into image restoration/enhancement method for video restoration/enhancement, their temporal modules are also limited by training-free manner. Recently, more and more amazing T2V model have emerged. These T2V models have been trained on a large amount of video data to learn powerful temporal priors which can preserve the temporal consistency of results. We propose to utilize the temporal priors in these T2V models to overcome the limitations of previous training-free temporal modules. Our method bridge the gap between image restoration/enhancement model and T2V model to achieve better results for video restoration/enhancement.

## 3 BACKGROUND

### 3.1 LATENT DIFFUSION MODELS

Latent diffusion models (LDM) operate in the latent space with VAE autoencoder $\mathcal{E}$, $\mathcal{D}$. First, an encoder $\mathcal{E}$ compresses the input RGB image/video $x$ to a low-resolution latent $z = \mathcal{E}(x)$, the forward and reverse diffusion process work on the latent, the latent can be reconstructed back to image/video $\mathcal{D}(z) \approx x$ by decoder $\mathcal{D}$. In the forward diffusion process, Gaussian noise is gradually added to $z_0$ to obtain $z_t$ through Markov transition with the transition probability

$$q(z_t|z_{t-1}) = \mathcal{N}(z_t; \sqrt{1-\beta_t}z_{t-1}, \beta_t \mathbf{I}) \tag{1}$$

where $\beta_t$ is the variance schedule for the timestep $t$. The backward process uses a trained U-Net $\varepsilon_\theta$ for denoising:

$$p_\theta(z_{t-1}|z_t) = \mathcal{N}(z_{t-1}; \mu_\theta(z_t, \tau, t), \Sigma_\theta(z_t, \tau, t)) \tag{2}$$

where $\tau$ denotes the textual prompt. $\mu_\theta$ and $\Sigma_\theta$ are computed by $\varepsilon_\theta$.

### 3.2 DDIM SAMPLING

DDIM sampling Song et al. (2020) is employed to reverse diffusion process, which converts noisy latent $z_T$ to a clean latent $z_0$ in a sequence of timestep:

$$z_{t-1} = \sqrt{\alpha_{t-1}} \frac{z_t - \sqrt{1-\alpha_t}\varepsilon_\theta}{\sqrt{\alpha_t}} + \sqrt{1 - \alpha_{t-1} - \sigma_t^2}\varepsilon_\theta + \sigma_t\epsilon_t \tag{3}$$

where $\alpha_t$ and $\sigma_t$ are parameters for noise scheduling Song et al. (2020), $\epsilon_t \sim \mathcal{N}(0, 1)$. In practice, firstly $\hat{z}_0$ is predicted from $z_t$

$$\hat{z}_0 = \frac{z_t}{\sqrt{\bar{\alpha}_t}} - \frac{\sqrt{1-\bar{\alpha}_t}\varepsilon_\theta}{\sqrt{\bar{\alpha}_t}} \tag{4}$$

where $\bar{\alpha}_t = \prod_{i=1}^{t} \alpha_i$. Then $z_{t-1}$ is sampled using both $\hat{z}_0$ and $z_t$

$$z_{t-1} = \frac{\sqrt{\alpha_t}(1-\bar{\alpha}_{t-1})}{1-\bar{\alpha}_t}z_t + \frac{\sqrt{\bar{\alpha}_{t-1}}\beta_t}{1-\bar{\alpha}_t}\hat{z}_0 + \sigma_t\epsilon_t \tag{5}$$

where $\beta_t = 1 - \alpha_t$.

### 3.3 Latent Diffusion Models for Zero-Shot IR

Linear inverse problem in image restoration (IR) can be formulated as

$$y = Ax + n \tag{6}$$

where $A$ is the linear degradation operator and $n$ is additive white Gaussian noise, the task is restoring the ground-truth image $x$ from the degraded image $y$. Following PSLD Rout et al. (2024), latent constraint is applied in the reverse diffusion process to preserve the content between generated result and degraded image. The constraint loss is formulated as

$$\mathcal{L} = \mathcal{L}_{rec} + \gamma_1 \mathcal{L}_{reg}$$
$$\mathcal{L}_{rec} = \|y - A(\mathcal{D}(\hat{z}_0))\|_2^2 \tag{7}$$
$$\mathcal{L}_{reg} = \|\hat{z}_0 - \mathcal{E}(A^T y + (I - A^T A)\mathcal{D}(\hat{z}_0))\|_2^2$$

where $\mathcal{L}_{rec}$ directly constrains the content, $\mathcal{L}_{reg}$ penalizes latents that are not fixed-points of the composition of the decoder-function with the encoder-function, make sure that the generated sample remains on the manifold of real data.

## 4 Method

Given a degraded video with $N$ frames $\{I_i\}_{i=0}^N$, our goal is to restore/enhance it to a clean/normal-light video $\{I_i''\}_{i=0}^N$. Our method leverages the homologous and heterogenous T2V models to assist in image restoration, achieving better temporal consistency and visual quality. We propose the corresponding homologous and heterogenous latents fusion, and further propose a COT-based fusion ratio strategy to update the fusion ratio self-adaptively. The framework is illustrated in Fig. 1. Our method is training-free and can be applied to any LDM-based image restoration methods, including zero-shot IR and trained IR method.

### 4.1 Homologous Latents Fusion

In the early stage of T2V model development, temporal modules are incorporated into the 2D denoising UNet from T2I LDM (Stable Diffusion) to construct the 3D denoising UNet for T2V LDM model, and the 2D VAE Encoder and Decoder from T2I are maintained for the T2V model. The representative T2V models are ModelScopeT2V Wang et al. (2023b) and ZeroScope Sterling (2023). For the Stable-Diffusion-based image restoration methods Rout et al. (2024); Lin et al. (2024), these T2V models share the same 2D VAE as those methods. We called this kind of T2V model the homologous T2V model. The latents between image restoration method and homologous T2V model can be directly fused, we called them homologous latents. We proposed to fuse the homologous latents between IR method and homologous T2V model to improve the temporal consistency.

To be specific, the image restoration/enhancement (IR/IE) model and homologous T2V model share the same initial noisy latents $z_T$, then the IR/IE model and homologous T2V model are applied to predict noise and generate the noisy latents $z_t^I$ and $z_t^{V1}$ by DDIM sampling, respectively. Different from T2V model, The IR/IE method predict the noise frame-by-frame and the denoised latents are concatenated along the temporal dimension. Then the noisy latents $z_t^I$ and $z_t^{V1}$ are fused to generate the fused noisy latents $z_t^{F1}$

$$z_t^{F1} = (1 - \lambda_t^{F1})z_t^I + \lambda_t^{F1} z_t^{V1} \tag{8}$$

where the hyper-parameter $\lambda_t^{F1}$ denoted the fusion weight. Then the IR/IE model and homologous T2V model share the fused noisy latents $z_t^{F1}$ and predict noise for the next denoising timestep.

### 4.2 Heterogenous Latents Fusion

Along with the development of T2V model, 2D VAE is replaced by 3D VAE, and the 3D denoising UNet is replace by MM-DiTs. The representative T2V models are CogVideoX Yang et al. (2024) and HunyuanVideo Kong et al. (2024), which have better consistency with the prompt, better temporal consistency and aesthetic quality than the previous T2V model. These T2V models have the different VAE from Stable-Diffusion-based image restoration methods, their latents can not be directly fused. We called this kind of T2V model the heterogenous T2V model. We further propose the heterogenous

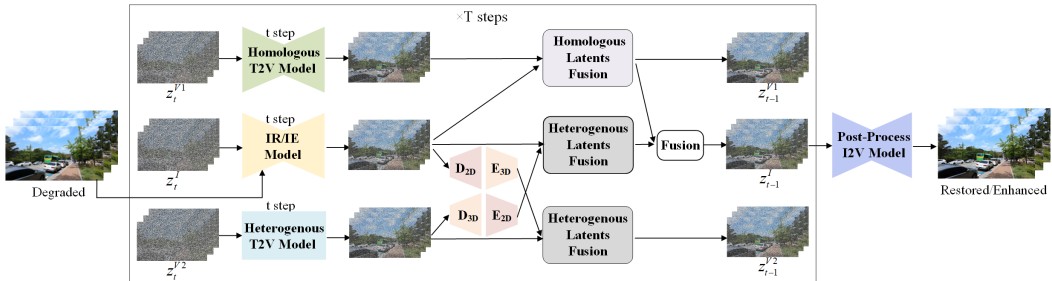

Figure 1: Overview of our framework.

latents fusion to utilize heterogenous T2V model to improve the temporal consistency and quality of video results.

Given the degraded video $\{I_i\}_{i=0}^N$, we encode it to latents $z_0^{V2}$ with the 3D VAE. Then add noise to the latents $z_0^{V2}$ to achieve initial noisy latents $z_T^{V2}$. Then we apply heterogenous T2V model to predict noise and generate the noisy latents $z_t^{V2}$ by DDIM sampling. The clean latents $\hat{z}_0^{V2}$ can be predicted from $z_t^{V2}$ by the Eq. 4. We decode the clean latents $\hat{z}_0^{V2}$ to video $\{I_i^{V2}\}_{i=0}^N$ by 3D VAE decoder $\mathcal{D}_{3D}$. Then we encode the video $\{I_i^{V2}\}_{i=0}^N$ to latents $\hat{z}_0^{V2\to I}$ by 2D VAE encoder $\mathcal{E}_{2D}$

$$\hat{z}_0^{V2\to I} = \mathcal{E}_{2D}(\mathcal{D}_{3D}(\hat{z}_0^{V2})) \tag{9}$$

According to Eq. 4, the noisy latents $z_t^{V2\to I}$ can be achieved by

$$z_t^{V2\to I} = z_t^{V2} + \sqrt{\bar{\alpha}_t^I}(\hat{z}_0^{V2\to I} - \hat{z}_0^{V2}) \tag{10}$$

where $\bar{\alpha}_t^I$ denotes the $\bar{\alpha}_t$ in the sampling of IR/IE model. Then the noisy latents $z_t^I$ and $z_t^{V2\to I}$ are fused to generate the fused noisy latents $z_t^{F2}$, and $z_t^{F2}$ is fused with $\lambda_t^F$ to get the final fused latents for the IR/IE model.

$$\begin{aligned}
z_t^{F2} &= (1 - \lambda_t^{F2})z_t^I + \lambda_t^{F2}z_t^{V2\to I} \\
z_t^F &= (1 - \lambda_t^F)z_t^{F1} + \lambda_t^F z_t^{F2}
\end{aligned} \tag{11}$$

where the hyper-parameter $\lambda_t^{F2}$ and $\lambda_t^F$ are the fusion weights. In a similar way, we can convert the IR/IE model latents to heterogenous T2V model latents to guide the sampling of heterogenous T2V model. To be specific, the clean latents $\hat{z}_0^I$ is predicted $z_t^I$ by the Eq. 4. We decode the clean latents $\hat{z}_0^I$ to video $\{I_i^I\}_{i=0}^N$ by 2D VAE decoder $\mathcal{D}_{2D}$ and encode it to latents $\hat{z}_0^{I\to V2}$ by 3D VAE encoder $\mathcal{E}_{3D}$

$$\hat{z}_0^{I\to V2} = \mathcal{E}_{3D}(\mathcal{D}_{2D}(\hat{z}_0^I)) \tag{12}$$

The noisy latents $z_t^{I\to V2}$ can be achieved by

$$z_t^{I\to V2} = z_t^I + \sqrt{\bar{\alpha}_t^{V2}}(\hat{z}_0^{I\to V2} - \hat{z}_0^I) \tag{13}$$

where $\bar{\alpha}_t^{V2}$ denotes the $\bar{\alpha}_t$ in the sampling of heterogenous T2V model. Then the noisy latents $z_t^{V2}$ and $z_t^{I\to V2}$ are fused to generate the final fused noisy latents $z_t^{FV2}$ for the heterogenous T2V model.

$$z_t^{FV2} = \lambda_t^{F2}z_t^{V2} + (1 - \lambda_t^{F2})z_t^{I\to V2} \tag{14}$$

Although the heterogeneous T2V model has better temporal consistency and video quality, heterogenous latents fusion alone is not optimal due to information loss during VAE encoding and decoding. Combining homologous and heterogenous latents through effective strategy can further improve the performance.

### 4.3 COT-BASED FUSION RATIO STRATEGY

How to set the fusion ratio $\lambda_t^{F1}$, $\lambda_t^{F2}$ and $\lambda_t^F$ at different timestep $t$ is a challenge problem. If $\lambda_t^{F1}$ and $\lambda_t^{F2}$ are too small, it would not work. Too large $\lambda_t^{F1}$ and $\lambda_t^{F2}$ will result in strong temporal consistent but blurry results or destroy the structure. Inspired by the development of chain-of-thought (COT) Lightman et al. (2023); Ma et al. (2023); Snell et al. (2024); Guo et al. (2025), we propose the COT-Based Fusion Ratio Strategy to solve this problem, which is shown in Fig. 2. We design a process reward model based on video quality as test-time verifiers within CoT reasoning paths.

Take $\lambda_t^{F1}$ as an example, we uniformly sample $M + 1$ values of $\lambda_T^{F1}$ from $[\lambda_t^c - r, \ \lambda_t^c + r]$ at timestep $t$. We calculate the different fused noisy latents $z_t^{F1}$ with different $\lambda_t^{F1}$, then predicted the corresponding clean latents $\hat{z}_t^{F1}$ by the Eq. 4. We decode each latents $\hat{z}_t^{F1}$ to a video $\{I_i^{F1}\}_{i=0}^N$ and calculate the average CLIP-IQA Wang et al. (2023a) and Wrap Error (WE) Lai et al. (2018) between all frames. The quality metrics CLIP-IQA and WE measure the visual quality and temporal temporal consistency of video, respectively. We rank all $M+1$ videos from 0 to $M$ (the lower the rank value, the better the quality) according to CLIP-IQA and WE, and obtain the rank values $R_{CLIP-IQA}$ and $R_{WE}$, respectively. We select the video with the lowest $(R_{CLIP-IQA}+R_{WE})$ as the video with best overall quality. And we select the corresponding $\lambda_t^{F1}$ value as the final $\lambda_t^{F1}$ at timestep $t$. Then set $\lambda_{t-1}^c$ to $\lambda_t^{F1}$ at timestep $t - 1$ and repeat this best-of-$N$ selection for $\lambda_{t-1}^{F1}$. The range $r$, sample number $M$, and initial $\lambda_T^{F1}$ are hyper-parameters. In a similar way, we calculate the different fused noisy latents $z_t^{F2}$ with different

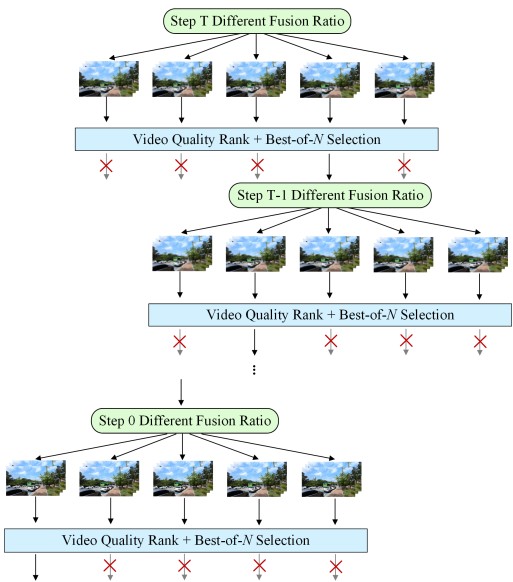

Figure 2: Our COT-Based Fusion Ratio Strategy. We design a process reward model based on video quality as test-time verifiers within CoT reasoning paths.

$\lambda_t^{F1}$ and select the final $\lambda_t^{F1}$. After the confirmation of $\lambda_t^{F1}$ and $\lambda_t^{F2}$, we select the final $\lambda_t^F$ through fused noisy latents $z_t^F$.

### 4.4 TEMPORAL-STRENGTHENING POST-PROCESSING

Fusing the latents between the IR/IE model and the T2V model can lead to better temporal consistency. However, the severe degree of temporal flickering of the IR model itself will still limit the performance. To solve this problem, we further propose to utilize the temporal prior in the I2V model Stable Video Diffusion (SVD) Blattmann et al. (2023) for temporal-strengthening post-processing. First, we encode the video $\{I_i'\}_{i=0}^N$ of IR/IE results (after latents fusion) into latents $z_0'$ with the VAE encoder of SVD. Since SVD is based on EDM sampling Karras et al. (2022), we invert $z_0'$ to noisy latents $z_T'$ with the inversion in the EDM framework:

$$z_{t+1}' = \frac{\sigma_{t+1} z_t' + (\sigma_t - \sigma_{t+1}) \, c_{\text{out}}^{t+1} \varepsilon_\theta' \left( c_{\text{in}}^t z_t'; c_{\text{noise}}^{t+1} \right)}{(\sigma_t - \sigma_{t+1}) \left( 1 - c_{\text{skip}}^{t+1} \right) + \sigma_{t+1}} \tag{15}$$

where $\sigma_t$, $c_{\text{skip}}^t$, $c_{\text{in}}^t$, $c_{\text{out}}^t$, and $c_{\text{noise}}^t$ are parameters for noise scheduling in EDM framework, $\varepsilon_\theta'$ is the SVD denoising network. Then we apply EDM sampling from $z_T''$ $(z_T')$ to $z_0''$:

$$z_t'' = z_{t+1}'' + \frac{\sigma_t - \sigma_{t+1}}{\sigma_{t+1}}$$
$$\left( z_{t+1}'' - \left( c_{\text{skip}}^{t+1} z_{t+1}'' + c_{\text{out}}^{t+1} \varepsilon_\theta' \left( c_{\text{in}}^{t+1} z_{t+1}''; c_{\text{noise}}^{t+1} \right) \right) \right) \tag{16}$$

When applying the SVD denoising network, the image condition is the first frame. Finally, we decode the latents $z_0''$ using the VAE decoder of SVD to obtain the final results $\{I_i''\}_{i=0}^N$. Through the reconstruction of SVD, video $\{I_i''\}_{i=0}^N$ has better temporal consistency than $\{I_i'\}_{i=0}^N$.

## 5 EXPERIMENTS

### 5.1 SETTINGS

For homologous and heterogeneous T2V models, we use ZeroScope and CogVideoX-2B, respectively. For the I2V model in post-processing, we use Stable Video Diffusion. We evaluate our method on two video restoration tasks (zero-shot video super-resolution and blind video super-resolution) and

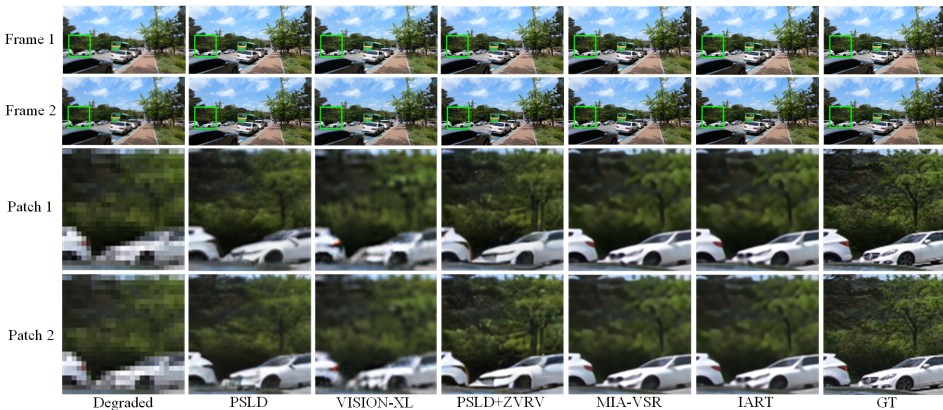

Figure 3: Visual quality comparison for zero-shot $4\times$ video super-resolution. Zoom in for better observation.

Table 1: Quantitative comparison with state-of-the-art methods for zero-shot $4\times$ video super-resolution. The best results are highlighted in bold and the second best results are underlined. The WE and t-LPIPS values have been multiplied by 100.

| Methods | Backbone | PSNR↑ | SSIM↑ | CLIP-IQA↑ | LPIPS↓ | WE↓ | FVD↓ | DOVER↑ | t-LPIPS↓ | VMAF↑ |
|---|---|---|---|---|---|---|---|---|---|---|
| TDAN(sup.) | - | 25.47 | 0.7359 | 0.3088 | 0.1869 | 0.5096 | 155.3 | 7.119 | 3.65 | 79.99 |
| BasicVSR++(sup.) | - | 28.12 | 0.8035 | 0.4823 | 0.1387 | 0.4683 | 122.5 | 7.849 | 1.84 | 82.75 |
| FMA-Net(sup.) | - | 28.29 | 0.8314 | 0.5121 | 0.1357 | 0.4923 | 117.6 | 8.207 | 1.06 | 84.56 |
| VRT(sup.) | - | 29.58 | 0.8464 | 0.5224 | 0.1221 | 0.3264 | 93.1 | 8.928 | 0.66 | 84.08 |
| MIA-SR(sup.) | - | 29.63 | 0.8466 | 0.5320 | 0.1219 | 0.3315 | 94.6 | 8.644 | 0.82 | 85.21 |
| IART(sup.) | - | 29.69 | 0.8472 | 0.5329 | 0.1252 | 0.3212 | 92.2 | 9.105 | 0.59 | 85.90 |
| PSLD | SDv1.5 | 26.80 | 0.7726 | 0.4315 | 0.1353 | 0.8408 | 140.4 | 4.783 | 6.28 | 75.80 |
| PSLD+Text2Video | SDv1.5 | 26.73 | 0.7619 | 0.4383 | 0.1419 | 0.8167 | 139.1 | 5.596 | 4.10 | 80.58 |
| PSLD+FateZero | SDv1.5 | 26.81 | 0.7852 | 0.4186 | 0.1475 | 0.7522 | 142.8 | 6.332 | 3.59 | 82.24 |
| PSLD+VidToMe | SDv1.5 | 26.85 | 0.7701 | 0.4423 | 0.1367 | 0.6649 | 138.3 | 6.560 | 2.64 | 81.06 |
| PSLD+FLDM | SDv1.5 | 26.89 | 0.7698 | 0.4519 | 0.1295 | 0.6051 | 130.2 | 6.937 | 2.37 | 82.63 |
| PSLD+ZVRV | SDv1.5 | 27.32 | 0.7845 | 0.5709 | 0.1110 | 0.2363 | 112.8 | 9.240 | 0.62 | 87.98 |
| PSLD | SDXL | 27.51 | 0.7802 | 0.4485 | 0.1361 | 0.8523 | 134.7 | 5.102 | 4.52 | 77.95 |
| VISION-XL | SDXL | 25.84 | 0.7554 | 0.2929 | 0.2487 | 0.4645 | 333.3 | 6.781 | 3.79 | 76.33 |
| PSLD+ZVRV | SDXL | 28.35 | 0.8099 | 0.5964 | 0.1072 | 0.2220 | 91.8 | 9.352 | 0.53 | 88.67 |

one video enhancement task (zero-shot low-light video enhancement). Following Cao et al. (2025), we collected 18 videos from REDS4, Vid4, and UDM10 for evaluation of zero-shot super-resolution, and collected 10 paired low-normal videos from the DID dataset Fu et al. (2023) for evaluation of zero-shot low-light video enhancement. Due to the slowly sampling speed and a test video contains a lot of frames, we crop and resize the frames to $576\times320$. For evaluation of blind super-resolution, we follow Yeh et al. (2024); Cao et al. (2025) and evaluate on DAVIS testing sets. For degraded videos of zero-shot video super-resolution, we follow the settings of linear degradation operator from Rout et al. (2024) and Fei et al. (2023). For blind super-resolution, low-quality videos are generated using the real-world degradation pipeline of RealBasicVSR Chan et al. (2022b). Due to page limitations, we provide comparisons on more tasks in the supplementary materials.

For zero-shot video enhancement, we utilize the same degradation model in Fei et al. (2023) and optimize the parameters of degradation model in the sampling process. The degradation model can be formulated as follows:

$$y = f\mathcal{D}(\hat{z}_0) + \mathcal{M} \tag{17}$$

where the factor $f$ is a scalar and the mask $\mathcal{M}$ is a matrix of the same dimension as $\mathcal{D}(\hat{z}_0)$. We optimize $f$ and $\mathcal{M}$ using gradient descent during the sampling process.

## 5.2 COMPARISON WITH STATE-OF-THE-ART METHODS

We utilize nine metrics to evaluate the restoration and enhancement quality. Besides the commonly used metrics PSNR, SSIM, we utilize CLIP-IQA and LPIPS to evaluate the visual perceptual quality, utilize Warping Error (WE) Lai et al. (2018), FVD Unterthiner et al. (2019), DOVER Wu et al.

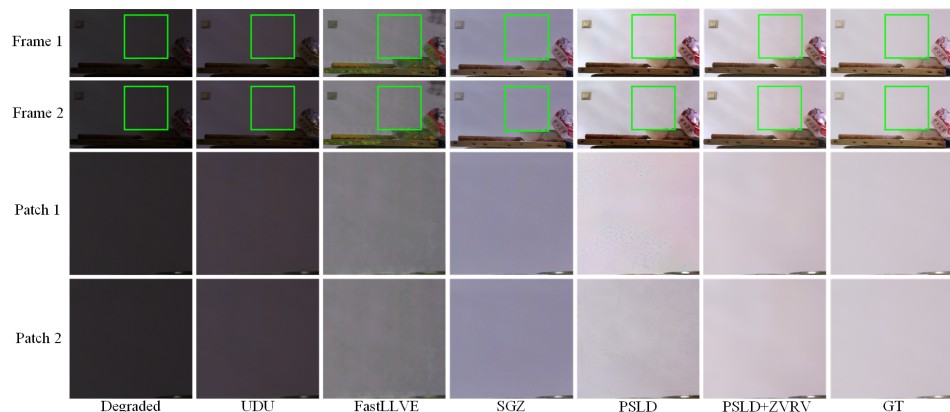

Figure 4: Visual quality comparison for zero-shot low-light video enhancement. Zoom in for better observation.

Table 2: Quantitative comparison with state-of-the-art methods for zero-shot low-light video enhancement. The best results are highlighted in bold and the second best results are underlined. The WE and t-LPIPS values have been multiplied by 100.

| Methods | Backbone | PSNR↑ | SSIM↑ | CLIP-IQA↑ | LPIPS↓ | WE↓ | FVD↓ | DOVER↑ | t-LPIPS↓ | VMAF↑ |
|---|---|---|---|---|---|---|---|---|---|---|
| FastLLVE (sup.) | - | 10.62 | 0.6911 | 0.2524 | 0.2674 | 0.1299 | 1465.2 | 2.093 | 15.76 | 49.33 |
| UDU (unsup.) | - | 6.24 | 0.4346 | 0.3560 | 0.3662 | 0.1048 | 1618.7 | 3.682 | 10.39 | 60.01 |
| SGZ | - | 15.03 | 0.6987 | 0.3835 | 0.1392 | 0.2454 | 723.6 | 5.715 | 1.10 | 81.81 |
| PSLD | SDv1.5 | 20.37 | 0.8281 | 0.3086 | 0.1186 | 0.6254 | 547.3 | 5.102 | 9.34 | 65.63 |
| PSLD+ZVRV | SDv1.5 | 20.69 | 0.8453 | 0.3955 | 0.0930 | 0.1017 | 251.9 | 8.965 | 0.78 | 83.22 |

(2023a), t-LPIPS Pérez-Pellitero et al. (2018); Chu et al. (2020), and VMAF Rassool (2017); Zheng et al. (2025) to evaluate the temporal consistency. In our supplementary file, we also provide the user study for human evaluation. For zero-shot video super-resolution, we compare our method with supervised (sup.) methods TDAN Tian et al. (2020), BasicVSR++ Chan et al. (2022a), FMA-Net Youk et al. (2024), VRT Liang et al. (2024), MIA-VSR Zhou et al. (2024b), IART Xu et al. (2024), and zero-shot video restoration method VISION-XL Kwon & Ye (2024). We also adding zero-shot video editing methods Text2Video-Zero Khachatryan et al. (2023), FateZero Qi et al. (2023), VidToMe Li et al. (2024), FLDM Lu et al. (2024) for comparison. For these zero-shot video editing methods, we transfer them to backbone PSLD. For zero-shot low-light video enhancement, we compare our method with supervised (sup.) method FastLLVE Li et al. (2023), unsupervised (unsup.) method UDU Zhu et al. (2024) and zero-shot video enhancement method SGZ Zheng & Gupta (2022). For blind video super-resolution, we compare our method with image super-resolution methods (DiffBIR Lin et al. (2024), DiT4SR Duan et al. (2025), TSD-SR Dong et al. (2025)) and video super-resolution methods (Upscale-A-Video Zhou et al. (2024a), SeedVR-7B Wang et al. (2025), DiffIR2VR Yeh et al. (2024), and ZVRD Cao et al. (2025)). DiffBIR, DiT4SR, and TSD-SR are also used as our backbones.

For a fair comparison, we change the stable diffusion backbone from Stable Diffusion v1.5 (SDv1.5) to Stable Diffusion XL (SDXL) when comparing with VISION-XL, which also uses SDXL as its backbone. Since the T2V model Zeroscope shares the same VAE encoder with SDv1.5, but the VAE encoder of SDXL is fine-tuned from that of SDv1.5, Zeroscope and SDXL do not use the same latent space. Thus, homologous latent fusion cannot be applied, we only apply the residual modules. Since DiT4SR is based on flow-matching sampling, which is inconsistent with the DDIM sampling used in T2V models ZeroScope and CogVideoX-2B, neither homologous nor heterogeneous latent fusion can be applied. Therefore, we only apply temporal-strengthening post-processing. For the one-step diffusion model TSD-SR, latent fusion is also inapplicable, only temporal-strengthening post-processing is employed. This also demonstrates that our method can be applied to any image-based model, regardless of its latent space or sampling method.

Table 1, 2, and 3 list the quantitative results on the evaluation data for zero-shot video super-resolution, zero-shot low-light video enhancement and blind video super-resolution, respectively. It

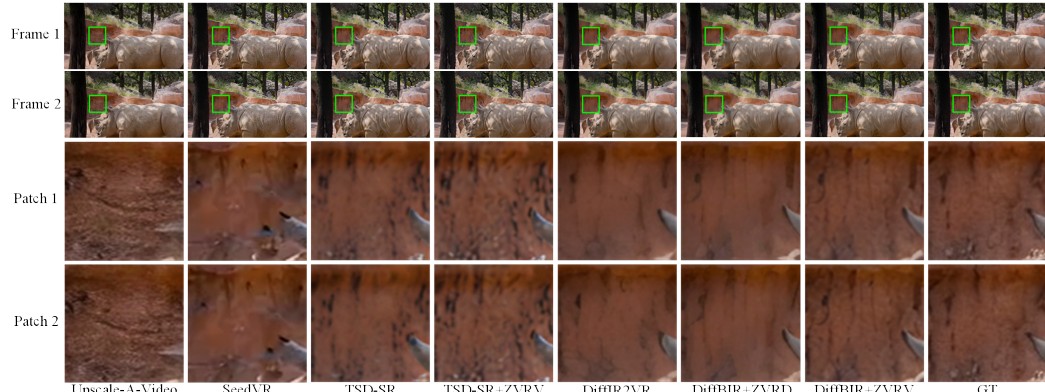

Figure 5: Visual quality comparison for 4× blind video super-resolution. Zoom in for better observation.

Table 3: Quantitative comparison with state-of-the-art methods for 4× blind video super-resolution on the DAVIS dataset. The best results are highlighted in bold and the second best results are underlined. The WE and t-LPIPS values have been multiplied by 100.

| Methods | Backbone | PSNR↑ | SSIM↑ | CLIP-IQA↑ | LPIPS↓ | WE↓ | FVD↓ | DOVER↑ | t-LPIPS↓ | VMAF↑ |
|---|---|---|---|---|---|---|---|---|---|---|
| Upscale-A-Video | SDv2.0 | 23.76 | 0.5676 | 0.7028 | 0.3114 | 0.5958 | 585.3 | 5.962 | 1.06 | 79.49 |
| SeedVR | SDv3.0 | 22.27 | 0.5162 | 0.8268 | 0.3070 | 0.5634 | 717.2 | 8.739 | 1.23 | 83.61 |
| DiT4SR | SDv3.0 | 23.41 | 0.6235 | 0.9283 | 0.2944 | 1.8274 | 917.6 | 4.305 | 11.29 | 73.24 |
| DiT4SR+ZVRV | SDv3.0 | 23.66 | 0.6387 | **0.9363** | 0.2909 | 0.5526 | 839.7 | 7.821 | 1.20 | 82.80 |
| TSD-SR | SDv3.0 | 25.15 | 0.6809 | 0.8702 | 0.1985 | 1.0295 | 357.8 | 6.464 | 2.12 | 77.93 |
| TSD-SR+ZVRV | SDv3.0 | 25.22 | 0.6866 | 0.8761 | 0.1811 | 0.4731 | 329.2 | 8.983 | 0.64 | 84.36 |
| DiffBIR | SDv2.1 | 26.50 | 0.6869 | 0.8340 | 0.1751 | 0.8061 | 279.1 | 5.720 | 3.92 | 76.64 |
| DiffIR2VR | SDv2.1 | 26.63 | 0.6904 | 0.8152 | 0.1743 | 0.7504 | 273.9 | 6.369 | 3.63 | 80.25 |
| DiffBIR+ZVRD | SDv2.1 | 26.86 | 0.7029 | 0.8017 | 0.1565 | 0.5042 | 262.5 | 7.551 | 0.55 | 81.02 |
| DiffBIR+ZVRV | SDv2.1 | **27.42** | **0.7388** | 0.8691 | **0.1395** | **0.3755** | **231.7** | **9.076** | **0.41** | **86.37** |

can be observed that our method ZVRV outperforms PSLD on nine metrics for both zero-shot tasks. For zero-shot 4× video super-resolution, our method outperforms PSLD (with SDXL backbone) with 0.84 dB gain for psnr, 0.1479 gain for CLIP-IQA, 4.25 gain for DOVER, 3.99 gain for t-LPIPS, 10.72 gain for VMAF and only has nearly 1/4 WE value. Our method also outperforms the supervised method IART on seven metrics. For zero-shot low-light video enhancement, our method outperforms PSLD on nine metrics and only has nearly 1/6 WE value. Our model can also be applied to blind video restoration by being inserted into blind image restoration in a zero-shot manner. For blind video super-resolution, our method with DiffBIR backbone achieves the best performance on eight metrics except CLIP-IQA, and our method with DiT4SR achieves the best performance on CLIP-IQA. Our method outperforms the SOTA method ZVRD with 0.56 dB gain for PSNR, 0.0359 gain for SSIM, 0.0674 gain for CLIP-IQA, 1.525 gain for DOVER, 5.35 gain for VMAF. In addition, our method, which utilizes the DiffBIR/TSD-SR backbone, significantly outperforms the SOTA methods Upscale-A-Video and SeedVR across all nine metrics. Table 4 lists the inference time of comparison method per frame at a resolution of 576×320. It can be observed that our method with the TSD-SR backbone has the fastest inference time.

Figs. 3, 4, and 5 present the visual comparison results on the evaluation data for zero-shot video super-resolution, zero-shot low-light video enhancement and blind video super-resolution, respectively. For video super-resolution, the areas of tree and car have different texture on the two frames of PSLD results. The results of VISION-XL, MIA-VSR, and IART are blurry. Our method can restore more sharp and temporal consistent textures. For low-light video enhancement, UDU, FastLLVE and SGZ have severe color shifts. There are color shift, different texture and brightness between two frames of PSLD. Our method preserve better temporal consistency. For blind video super-resolution, our method can also restore more temporally consistent textures on the rock, which is also closest to the ground truth. The result of Upscale-A-Video has artifacts, the result of SeedVR is blurry, and neither is faithful to the ground truth.

Table 4: Inference Time on GPU 96G H20.

| Methods | Upscale-A-Video | SeedVR | DiffBIR+ZVRV | DiT4SR+ZVRV | TSD-SR+ZVRV |
|---|---|---|---|---|---|
| Time (s) ↓ | 4.12 | 10.55 | 61.29 | 21.83 | 2.96 |

Due to page limitations, we provide more comparisons and a demo video in the supplementary materials.

## 5.3 ABLATION STUDY

In this section, we perform an ablation study to demonstrate the effectiveness of the proposed Homologous Latents Fusion, Heterogenous Latents Fusion, COT-Based Fusion Ratio Strategy, and Temporal-Strengthening Post-Processing. Besides the nine metrics, we also list the runtime per frame at a resolution of 576×320, the peak GPU memory usage, and the number of parameters when each module is added. Taking 4× blind video super-resolution as an example, Table 5 lists the quantitative comparison results in the evaluation data by adding these modules one by one. When the COT-Based Fusion Ratio Strategy is not added, Homologous and Heterogenous Latents Fusion use a simple linear update schedule. It can be observed that both Homologous and Heterogenous latents can significantly improve the temporal consistency. Homologous Latents Fusion can bring 0.19 dB gain for PSNR, 0.0112 gain for SSIM, and 0.1104 gain for WE. Heterogenous Latents Fusion can bring 0.43 dB gain for PSNR, 0.022 gain for SSIM, 0.1421 gain for WE, and 18.2 gain for FVD. However, Homologous Latents Fusion results in an increase of

Table 5: Ablation study for Homologous Latents Fusion (HMLF), Heterogenous Latents Fusion (HTLF), COT-Based Fusion Ratio Strategy (CFRS) and Temporal-Strengthening Post-Processing (TSPP) on 4× blind video super-resolution task. The WE and t-LPIPS values have been multiplied by 100.

| | | | | | |
|---|---|---|---|---|---|
| HMLF | × | ✓ | ✓ | ✓ | ✓ |
| HTLF | × | × | ✓ | ✓ | ✓ |
| CFRS | × | × | × | ✓ | ✓ |
| TSPP | × | × | × | × | ✓ |
| PSNR↑ | 26.50 | 26.69 | 27.12 | 27.35 | 27.42 |
| SSIM↑ | 0.6869 | 0.6981 | 0.7201 | 0.7256 | 0.7388 |
| CLIP-IQA↑ | 0.8340 | 0.7339 | 0.7026 | 0.8446 | 0.8691 |
| LPIPS↓ | 0.1751 | 0.1702 | 0.1678 | 0.1425 | 0.1395 |
| WE↓ | 0.8061 | 0.6957 | 0.5536 | 0.4942 | 0.3755 |
| FVD↓ | 279.1 | 273.8 | 255.6 | 248.2 | 231.7 |
| DOVER↑ | 5.720 | 6.005 | 7.028 | 7.459 | 9.076 |
| t-LPIPS↓ | 3.92 | 3.36 | 1.98 | 1.63 | 0.41 |
| VMAF↑ | 76.64 | 78.35 | 80.96 | 82.20 | 86.37 |
| Time(s)↓ | 9.15 | 22.72 | 51.98 | 58.69 | 61.29 |
| Memory(GB)↓ | 8.3 | 16.0 | 54.2 | 54.2 | 72.8 |
| Params(B)↓ | 1.4 | 3.1 | 5.1 | 5.1 | 7.4 |

13.57 seconds in runtime, 7.7 GB in peak GPU memory, and 1.7B in parameters, Heterogenous Latents Fusion results in an increase of 29.26 seconds in runtime, 38.2 GB in peak GPU memory, and 2B in parameters. The COT-Based Fusion Ratio Strategy can significantly improve the value of CLIP-IQA and LPIPS. It also brings 0.23 dB gain for PSNR, and 0.0594 gain for WE. And it results in an increase of 6.71 seconds in runtime. Temporal-Strengthening Post-Processing can further reduce the WE and FVD value. It only adds 2.65 seconds in runtime, but results in an increase of 18.6 GB in peak GPU memory usage and 2.3B in parameters. Our method with DiffBIR backbone, has 7.4B parameters but achieves significantly better performance than SeedVR-7B, which has 7B parameters.

## 6 CONCLUSION

In this paper, we propose the first framework for for zero-shot video restoration and enhancement with assistance of text-to-video diffusion model. Through the proposed homologous and heterogenous latents fusion, we can utilize any kind of SOTA T2V model to assist the image restoration/enhancement model in achieving temporal consistent video restoration/enhancement. We further propose the COT-based fusion ratio strategy to better control the fusion ratio when fusion latents at each timestep. Experimental results demonstrate the superiority of the proposed method in performance and temporal consistency.

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

## A APPENDIX

### A.1 USE OF LLMS

We did not use Large Language Models (LLMs) in paper writing at all.

