# OpenReview forum: "Zero-Shot Video Restoration and Enhancement with Assistance of Video Diffusion Models"
_ICLR.cc/2026/Conference — Submitted to ICLR 2026_

### Official Review · Reviewer_jDUK · 2025-10-28

**Soundness:** 4
**Presentation:** 4
**Contribution:** 4
**Rating:** 8
**Confidence:** 4

**Summary:**

This paper proposes ZVRV, a training-free framework that leverages video diffusion models to improve temporal consistency in zero-shot video restoration. Key contributions include homologous and heterogeneous latent fusion methods (enabling the use of any T2V model regardless of VAE compatibility), a COT-based strategy for adaptive fusion-weight selection, and temporal post-processing with I2V models. Experiments show 73-84% reduction in temporal flickering across super-resolution and low-light enhancement tasks, achieving state-of-the-art results.

**Strengths:**

- The paper identifies and addresses a real limitation: temporal flickering when applying image restoration methods to videos by leveraging temporal priors from pre-trained video diffusion models.
- The key technical insight enabling cross-VAE latent fusion through encode-decode operations is elegant and practical. This allows the framework to utilize state-of-the-art T2V models (CogVideoX, HunyuanVideo) regardless of VAE compatibility.
- Demonstrates 73-84% reduction in temporal flickering (Warping Error) while maintaining/improving visual quality metrics. The method outperforms both supervised and zero-shot baselines across three different tasks.
- The framework requires no training and works with any LDM-based restoration method, making it immediately practical and deployable.
- Applying chain-of-thought reasoning with test-time verification to select fusion ratios is an interesting adaptation of LLM techniques to diffusion models.

**Weaknesses:**

- The method runs multiple models simultaneously (image restoration + homologous T2V + heterogeneous T2V + I2V post-processing), and the COT strategy requires M+1 forward passes at each timestep for three different fusion ratios (\lambda^F1, \lambda^F2, \lambda^F). This represents massive computational overhead, yet the paper provides no runtime analysis, memory requirements, or comparison with baselines.
- The paper's core contributions have limited novelty. Homologous fusion is acknowledged as similar to FVDM (Lu et al. 2024), adapted from video editing to restoration, and heterogeneous fusion via VAE encode-decode is relatively straightforward. The COT application to fusion ratio selection, while interesting, feels more like an engineering trick than a fundamental contribution. The central insight is essentially "use video diffusion models to reduce temporal flickering," which is intuitive but not deeply novel.
- The evaluation uses very small test sets (18 videos for super-resolution, 10 for enhancement), raising concerns about generalization, and resolution is severely restricted to 576x320 due to computational constraints, which doesn't reflect real-world high-resolution video restoration needs. The paper lacks failure case analysis or discussion of when the method doesn't work, and critical ablations are missing for key hyperparameters such as M and r in the COT strategy, and the choice of fusion schedules.

**Questions:**

- Can you provide detailed runtime comparisons (wall-clock time, memory usage, FLOPs) against baseline methods? How many total forward passes does your method require per video with typical M values? What is the practical maximum resolution your method can handle on standard GPUs, and how does this scale?
- How sensitive is performance to the choice of M (sample number) and r (range)? Can you provide ablation studies showing the performance vs. computational cost trade-offs for different M values? Why use COT sampling at every timestep rather than learning or predicting good fusion ratios?
- The test sets are very small (18 and 10 videos). Have you evaluated on larger benchmarks? Can you provide results on standard video restoration datasets at full resolution? How does the method perform on longer videos (>50 frames)?

---

> ### Author Response · Authors · 2025-12-03
>
> R4-Q1: Thanks! We have added the ablation study for runtime and peak GPU memory in Tab. 5 of the main paper. We also provide the run time comparisons with SOTA methods in Tab. 4. The run time of our method depends on the image restoration backbone. A lighter backbone leads to smaller inference time.
>
> R4-Q2: We would like to point out that our method is the first approach that utilizes video diffusion model to assist any image diffusion model for zero-shot video restoration. This is not straightforward. It is easy for homologous fusion when the video diffusion models share the same latent space with image diffusion models. However, video diffusion is developing fast, and the modern video diffusion models did not share the latent space with the image model. To solve this problem, we propose the heterogeneous fusion strategy. However, when the T2V video diffusion model and the image diffusion model utilize different sampling methods (such as flow-matching and DDIM), or when the image diffusion model is a one-step diffusion model, latent fusion cannot be applied. To solve this problem, we further propose a temporal-strengthening post-processing method. In this way, our method can be applied to any image diffusion model. In addition, we make this strategy work well for many video restoration tasks, outperforming SOTA methods by a large margin. We would like to point out that many classical methods in deep learning era are not proposing a new concept but providing an effective method by combining different strategies, such as DiT (Diffusion+Transformer), which are also important for the development of AI models.
>
> R4-Q3,R4-Q5: For testing datasets, we follow the settings in ZVRD (2025-AAAI). The test frames for zero-shot video restoration/enhancement are also down-cropped to 256×256 in ZVRD. It is worth noting that we do not downsample but only crop the input frames in blind video SR to avoid influencing the degradation. Cropping patches for testing is a common strategy in video restoration tasks. In addition, our method can divide the frame into patches and use aggregation sampling in StableSR for higher resolution. In the supplementary material, we have added an ablation study on the key hyperparameters (M and r) in the COT strategy and on the choice of fusion schedules.
>
> R4-Q4: We have added the ablation study for runtime and peak GPU memory in Table 5 of the main paper. Since we apply the strategy every 10 timesteps and then apply the obtained 3 hyper-parameters to the following 9 timesteps, there are 12(M+1) forward passes of SDv1.5/SDv2.1 VAE encoder for 50-step DiffBIR.  For high-resolution input, we divide the frame into patches and use aggregation sampling in StableSR. In this way, our method can deal with inputs with arbitrary resolutions.
>
> R4-Q6: Cropping patches for testing is a common strategy in video restoration tasks. We will provide results at full resolution in the final version. For long video, we separate it into different video clips, neighboring video clips share one overlapping frame. When our Temporal-Strengthening Post-Processing finishes the process of the previous video clip, the last/shared frame is used as the image condition in the next video clip to maintain the long-range temporal consistency. There is no limitation for the length of the test video.

---

### Official Review · Reviewer_FoHs · 2025-10-31

**Soundness:** 3
**Presentation:** 3
**Contribution:** 3
**Rating:** 4
**Confidence:** 4

**Summary:**

This paper proposes a zero-shot video restoration framework that leverages diffusion generative priors for multiple degradation tasks without training. The method performs diffusion inversion on each frame and introduces a temporal alignment module for consistency across frames. The approach is applied to denoising, deblurring, and super-resolution tasks, showing competitive zero-shot performance.

**Strengths:**

- The proposed zero-shot setting is novel and practically motivated.
- Integrating diffusion priors with temporal alignment is conceptually elegant.
- The method demonstrates versatility across multiple restoration tasks.

**Weaknesses:**

- Lacks comparison with recent diffusion-based video restoration models such as **Upscale-A-Video (CVPR 2024)** and **SeedVR (CVPR 2025)**, making it hard to gauge true competitiveness.
- No runtime, peak memory, or parameter analysis is provided, which limits understanding of efficiency and scalability.
• Temporal consistency evaluation is weak, reporting only **Warping Error (WE)** without metrics like **DOVER** or **tLPIPS**, which better reflect human-perceived temporal coherence and detail stability.

**Questions:**

1. How does the method perform compared with diffusion-based video restoration baselines such as Upscale-A-Video or SeedVR
2. Can the authors report runtime, peak memory, and parameter counts for a fair efficiency analysis
3. Please include more temporal consistency metrics (e.g., DOVER, tLPIPS) to strengthen the evaluation

---

> ### Author Response · Authors · 2025-12-03
>
> R3-Q1,R3-Q4: We have added quantitative comparisons with Upscale-A-Video and SeedVR in Tables 3 and 4, and we have included the corresponding visual quality comparison in Figure 5. Our method outperforms the two methods by a large margin.
>
> R3-Q2,R3-Q5: We have added the ablation study for runtime, peak GPU memory, and parameter in Table 5 of the main paper.
>
> R3-Q3,R3-Q6: We have added DOVER and t-LPIPS metrics for all quantitative comparisons. Our method still achieves the best performance in terms of the two new metrics.

---

### Official Review · Reviewer_jDRk · 2025-11-01

**Soundness:** 3
**Presentation:** 3
**Contribution:** 2
**Rating:** 4
**Confidence:** 4

**Summary:**

I think the paper grafts pre-trained T2V/I2V diffusion models onto image-based zero-shot IR to reduce flicker:  (1) homologous latent fusion when the 2D VAE matches.  (2) heterogeneous latent fusion via 2D↔3D VAE round-trips. (3) a “CoT-based” per-timestep fusion-ratio search with CLIP-IQA + warping-error as the verifier.  (4) A Stable Video Diffusion post-processing pass for extra temporal smoothing. On REDS/Vid4/UDM10/DAVIS and a small low-light set, the method beats PSLD and some zero-shot/editing baselines on PSNR/SSIM/LPIPS/CLIP-IQA/WE/FVD.

**Strengths:**

(1) I think the goal is practical: using actual video priors to fix temporal instability in zero-shot diffusion IR.

(2) The heterogeneous latent bridge (2D↔3D VAE encode/decode to align latents) is a straightforward engineering workaround that makes modern T2V usable.

(3) The pipeline is training-free w.r.t. new networks and slots into several zero-shot IR backbones.

(4) Results show lower WE/FVD and perceptual gains over PSLD-only variants; ablations indicate each block helps.

**Weaknesses:**

(1) “First framework” is oversold. I think the novelty is thinner than claimed. Homologous fusion is basically FVDM-style latent mixing applied to restoration rather than editing; heterogeneous fusion is a vanilla encode–decode bridge between VAEs; and the “CoT-based” strategy is just a best-of-N hyperparameter search per timestep with two off-the-shelf metrics. Slapping “CoT” on a verifier-guided grid search doesn’t make it reasoning-based. The pitch feels buzzwordy rather than conceptually new.

(2) The “training-free” recipe quietly burns a lot of test-time compute. I think sampling multiple fusion ratios at every diffusion step for every clip is a huge multiplier on runtime. Then you post-process with SVD (inversion + EDM sampling) on top. There’s no honest wall-clock accounting or energy/latency comparison vs. stronger baselines at their own optimal step counts. This makes the method look practical on paper but painful in reality.

(3) Metric gaming and small-scale evaluation. The main verifier metric at test-time includes CLIP-IQA, which your own method optimizes against during ratio selection; unsurprisingly, you win it. Datasets are tiny (e.g., 18 videos for SR; 10 for low-light). Frames are even down-cropped to 576×320 “due to slow sampling, which conveniently hides high-res temporal artifacts. I think this undercuts the generality of the claims.

(4) Fairness & clarity of baselines. I think comparisons are muddy: you mix supervised VSR models (trained for the task) with zero-shot methods, and for diffusion baselines, you lock them to specific step counts instead of reporting speed–quality curves. The DavIS blind SR table shows large gains, but I don’t see solid controls that exclude metric bias from the verifier loop.

**Questions:**

- What’s the end-to-end wall-clock (and GPU budget) per 1-second 1080p clip for your full pipeline (fusion-ratio search + SVD)? Give a step-wise breakdown.

- Verifier leakage: Since CLIP-IQA is used to select fusion ratios, do you also report metrics not used in selection (e.g., t-LPIPS variants, VMAF-temporal) where you didn’t tune?

- Please provide speed–quality trade-off curves for PSLD/other diffusion baselines (10/25/50 steps) and your method with/without SVD, at matched wall-clock.

---

> ### Author Response · Authors · 2025-12-03
>
> R2-Q1: We agree with you that our method is not a totally new concept. However, our method is the first approach that utilizes video diffusion model to assist any image diffusion model for zero-shot video restoration. This is not straightforward. It is easy for homologous fusion when the video diffusion models share the same latent space with image diffusion models. However, video diffusion is developing fast, and the modern video diffusion models did not share the latent space with the image model. To solve this problem, we propose the heterogeneous fusion strategy. However, when the T2V video diffusion model and the image diffusion model utilize different sampling methods (such as flow-matching and DDIM), or when the image diffusion model is a one-step diffusion model, latent fusion cannot be applied. To solve this problem, we propose a temporal-strengthening post-processing method. In this way, our method can be applied to any image diffusion model. In addition, we make this strategy work well for many video restoration tasks, outperforming SOTA methods by a large margin. We would like to point out that many classical methods in deep learning era are not proposing a new concept but providing an effective method by combining different strategies, such as DiT (Diffusion+Transformer), which are also important for the development of AI models.
>
> R2-Q2: All zero-shot based methods involve large computation cost during the inference. We have added computational comparisons and computational ablation studies in Tables 4 and 5 of the main paper.
>
> R2-Q3,R2-Q6: Although we optimize the CLIP-IQA and WE metrics during the ratio selection process, we also report metrics that were not used in the selection process (e.g., PSNR, SSIM, LPIPS, FVD, DOVER, t-LPIPS, and VMAF), which were not tuned. For testing datasets, we follow the settings in ZVRD (2025-AAAI). The test frames for zero-shot video restoration/enhancement are also down-cropped to 256×256 in ZVRD. It is worth noting that we do not downsample but only crop the input frames in blind video SR to avoid influencing the degradation. Cropping patches for testing is a common strategy in video restoration tasks. We did not observe high-resolution temporal artifacts for this case.
>
> R2-Q4,R2-Q7: For all diffusion baselines, we report their performance using the number of diffusion steps in their official code, which corresponds to their best performance. This is also a common practice in most papers (Upscale-A-Video, SeedVR, ZVRD, etc). We also report their corresponding speed in Table 4 of the main paper. We also report the speed of our method with and without SVD (temporal-strengthening post-processing) in Table 5 of the main paper.
>
> R2-Q5: We conducted ablation studies on runtime per frame and GPU budget at a resolution of 576×320, as shown in Table 5 of the main paper, and performed a runtime comparison in Table 4. For higher resolutions, we divide the frame into patches and apply aggregation sampling in StableSR. Under this approach, the GPU budget does not increase, and the inference time for a 1-second 1080p clip at 25 fps would be 400 times of our current reported inference time.

---

### Official Review · Reviewer_vAKE · 2025-11-01

**Soundness:** 2
**Presentation:** 2
**Contribution:** 2
**Rating:** 6
**Confidence:** 2

**Summary:**

The paper proposes a new video restoration method that leverages an existing image restoration model, along with homologous and heterogeneous T2V models. The key idea is to fuse the latents from different models to perform per-frame restoration while maintaining temporal consistency. A post-processing step is then applied using an I2V model to further enforce consistency. The method generally shows improved results when combined with an existing image restoration (IR) method.

**Strengths:**

1. The method is unsupervised and training-free, leveraging existing pre-trained models, which makes it practical.

2. The method generally shows improvements compared to the baseline or other compared methods.

**Weaknesses:**

1. I think the method section could be much better presented and many details should be introduced. I struggled to fully understand the method.  How do you get the noise $z_T$ in line 201? Do you invert the degraded video? How do you use the T2V model to generate a video similar to the input one?  Which text prompt are you using? In line 182, you mentioned that your input is only a video.

2. Based on my understanding, I think the performance depends heavily on the hyperparameters used for latent fusion, which makes the method seem somehow ad hoc.

3. The method applies multiple models, which makes it computationally expensive. A computational comparison is needed.

**Questions:**

See weaknesses.

---

> ### Author Response · Authors · 2025-12-03
>
> R1-Q1: We utilize different $z_T$ settings  according to the settings of the backbone. For the PSLD backbone, we invert the degraded video to obtain noisy latents, $z_T$. For the DiffBIR backbone, the noisy latents $z_T$ are Gaussian noise. For the T2V model, we utilize null prompt. To make the T2V model generate a video similar to the input, we utilize latent fusion. The latent fed to the T2V model is a combination of the image restoration latent, which contains the image content, and T2V latent. In this way, the T2V model can generate temporal consistent contents which are similar to the degraded input. In addition, when we utilize PSLD backbone, the latent input to the T2V model is also the inverted degraded video latent, which also makes the T2V model generate content similar to the degraded input.  We will add more details to make our settings and method clear in the final version.
>
> R1-Q2: Yes, the performance is dependent on the hyperparameters for latent fusion. However, we did not manually set these parameters. In contrast, we propose a COT-Based Fusion Ratio Strategy to identify the optimal hyperparameters. In this way, these hyperparameters are automatically set. Therefore, our method is a general solution other than an ad hoc solution.
>
> R1-Q3: We have added computational comparisons and computational ablation studies in Tables 4 and 5 of the main paper. We would like to point out that our computation cost heavily depends on the backbone. When we utilize the efficient TSD-SR as our image restoration backbone, our inference time is 2.96 s, which is much smaller than the video model (Upscale-A-Video), which is 4.12 s. When we use DiffBIR as the backbone, our inference time is larger, and Tab. 5 presents the step-by-step inference time for our full model.

---

### Meta-Review · Area_Chair_k3Zq · 2025-12-24

**Summary:**

This paper receives two marginally below the acceptance threshold, one marginally above the acceptance threshold and one accept. After the rebuttal, I still believe there several major issues not addressed including: (1) unclear presentation (vAKE), (2) high compuational costs (vAKE jDRk FoHs jDUK), (3) over-sold novelty (jDRk jDUK) and (4) comparison issues in experiments (jDRk FoHs jDUK). As a result, this paper cannot be accepted.

**Reviewer Scores:**

NA

---

### Decision · Program_Chairs · 2026-01-26

Reject